# Community-Acquired and Healthcare-Associated Sepsis: Characteristics and in-Hospital Mortality in Italy

**DOI:** 10.3390/antibiotics9050263

**Published:** 2020-05-19

**Authors:** Gabriella Di Giuseppe, Maria Mitidieri, Federica Cantore, Concetta P. Pelullo, Maria Pavia

**Affiliations:** Department of Experimental Medicine, University of Campania “Luigi Vanvitelli”, 81100 Naples, Italy; gabriella.digiuseppe@unicampania.it (G.D.G.); maria.mitidieri@unicampania.it (M.M.); fedcantore@libero.it (F.C.); concettapaola.pelullo@unicampania.it (C.P.P.)

**Keywords:** community-acquired sepsis, healthcare-associated sepsis, mortality, sepsis

## Abstract

*Background***:** The main aim of the study was to analyse characteristics of sepsis according to the setting of occurrence and to identify predictors of sepsis-related in-hospital mortality. *Methods***:** 544 medical records of adult patients with a diagnosis of sepsis were consulted and divided into two groups according to the setting where sepsis originated: community-acquired (CA) and healthcare-associated (HA) sepsis. *Results***:** Overall, 257 (47.2%) patients had HA sepsis and the in-hospital death rate was 33.6%. Results of the multiple logistic regression revealed that patients with HA sepsis were significantly more likely to have been admitted from another hospital or ward, to have a ≥1 Charlson’s index, to be immunesuppressed, and to have undergone a surgical intervention during hospitalization. In-hospital deaths were significantly associated with older age, admission from another hospital or ward, need of haemodialysis and mechanical ventilation (MV), whereas they were less likely in patients with HA sepsis as compared with CA sepsis. *Conclusion*: Community-acquired and HA sepsis show distinct clinical, prognostic and risk factors profiles, and should be managed according to their differential characteristics.

## 1. Introduction

It is well known that sepsis, recently defined as a life-threatening organ dysfunction caused by a dysregulated host response to infection [1], remains an unresolved and challenging public health issue and, although its true incidence and mortality are difficult to calculate, it is one of the leading causes of death globally. 

Researchers and clinicians have long debated on the need to improve prevention, early detection, and clinical management of sepsis, and in 2017 the World Health Assembly and WHO have strongly focused attention on several sepsis-related issues and have declared sepsis a global health priority [2]. 

Recognition of sepsis is far from straightforward, due to the vague and non-specific presentation in the early stages and to the extremely variable clinical characteristics; therefore the identification of markers for early detection that predict the development of sepsis and associated mortality is still challenging, since sepsis outcomes are largely dependent from prompt and effective management through control of underlying infection, support of organs dysfunctions, and resuscitation. 

Several patients’ characteristics are associated with a higher risk of developing sepsis; indeed elderly patients, affected by severe comorbidities, such as cancer or renal failure, as well as by impaired immunity, including stressors such as surgery, trauma or burns have been found to have a significantly higher risk of developing sepsis [3]. 

Recently, patients with sepsis have been classified in clinically distinguished subgroups according to the setting where sepsis originated: (1) patients admitted with a “community-acquired” (CA) sepsis; (2) those who develop sepsis few days after admission and recognize one or more “healthcare-associated” risk factors prior to admission; and (3) those who are diagnosed a “hospital-acquired” sepsis during hospital stay. It has been reported that it may be misleading to consider these entities as a whole group, since they may be very dissimilar as regards to several characteristics such as aetiology, risk factors, underlying infection, patients’ characteristics, onset, and prognosis [4]; therefore, one of the challenges in the description of the burden and mortality related to sepsis is the opportunity to separately consider these diverse clinical entities. 

Numerous studies have described the epidemiology of healthcare-associated infections (HAIs) in Italy [5,6,7,8,9,10], whilst little data are available about sepsis [3], although a recent study aimed to estimate the nationwide burden and time trends of sepsis-related mortality in Italy, found that the number of death certificates reporting sepsis increased from 18,939 in 2003 to 49,010 in 2015, for an increase from 3% to 8% of all deaths [11]. Therefore, the main aim of the study was to analyse characteristics of sepsis according to the setting of occurrence, and to identify predictors of sepsis-related in-hospital mortality.

## 2. Results

Of the 42 selected hospitals, 15 agreed to participate (response rate 35.7%) and 544 clinical records were available for inclusion in the study. Among the enrolled hospitals, 12 were general, one a teaching and two were specialty hospitals, respectively devoted to cancer and infectious diseases research and healthcare. Number of beds ranged from 90 to 377. No substantial differences were found between participating and non-participating hospitals. 

### 2.1. Patients’ Characteristics

The main characteristics of the included patients are listed in Table 1. Patients were equally distributed by gender, with a mean age of 62.8 years (interquartile range 51–76), and a mean Charlson’s et al. [12] comorbidity index of 3.3 (interquartile range 1–5). The great majority of patients had at least one comorbidity; in particular, 280 (51.5%) had cardiovascular disease, 140 (25.7%) diabetes mellitus, 146 (26.8%) cancer, and 109 (20%) chronic obstructive pulmonary disease (COPD); moreover, 109 (20%) were immune suppressed. 

### 2.2. Clinical Outcomes

The main characteristics of underlying infections (sites and involved micro-organisms) are reported in Table 2. The most common infections associated to sepsis were gastrointestinal (24.6%), bloodstream (20.4%) and respiratory tract infections (17.8%), and the isolated micro-organisms, mainly from urine (20.8%) and blood cultures (40.4%), were *Staphylococcus* spp. (24.8%), *Candida* spp. (17.3%), *Escherichia coli* (15.8%), *Klebsiella* spp. (11.8%), and *Acinetobacter* spp. (11.5%). 

Characteristics of patients’ clinical course and outcomes are reported in Table 3. The mean length of stay was 20.8 days (range 1–300), 167 (30.7%) had been admitted from another hospital/health facility and 43 (7.9%) from another ward. One forth was admitted in intensive care units (ICUs) and 26.3% underwent surgical interventions during hospitalization. Mechanical ventilation (MV) was provided to 40.1%, haemodialysis to 13.4%, and ICU was needed during hospitalization by 17.3% of patients. Overall, 257 (47.2%) patients had healthcare-associated (HA) sepsis, and one-third (33.6%) died during hospitalization. 

### 2.3. Univariate Analysis

In univariate analysis, compared to CA sepsis, patients with HA sepsis were significantly more likely to have been admitted from another hospital or ward (49.4% versus 28.9%), to have been admitted in ICU (31.5% versus 19.1%) and in surgical wards (14.4% versus 7%), to have a ≥1 Charlson’s index (90.7% versus 83.6%), to have been diagnosed a cancer (37.6% versus 17.4%), to be immune suppressed (24.9% versus 15.7%), to have undergone a surgical intervention during hospitalization (40.5% versus 13.6%) and to have needed MV (46.3% versus 34.5%). Moreover, in patients with HA sepsis a gastrointestinal primary site of infection was significantly more frequent (30% versus 19.9%), as well as isolation from body fluids of *Candida* spp. (22.2% versus 12.9%), *Acinetobacter* spp. (14.8% versus 8.7%) and *Pseudomonas* spp. (10.9% versus 3.2%). On the contrary, patients with CA sepsis were significantly more likely to be diabetics (29.6% versus 21.4%) and to have a genitourinary (14.3% versus 8.2%) or central nervous system (10.8% versus 3.9%) or unknown/undetermined (10.1% versus 3.5%) primary site of infection (Table 1, Table 2 and Table 3). 

In univariate analysis in-hospital death was significantly more frequent in older patients, in those admitted from another hospital or ward (61.8% versus 26.9%), in those admitted in ICU (51.9% vs. 11.4%), in patients needing haemodialysis (27.3% versus 6.4%), MV (82.5% versus 18.6%) or having undergone clean-contaminated surgical interventions during hospitalization (81.8% versus 59.1%). Primary sites of infections significantly predicting in-hospital death were pulmonary (33.3% versus 10%) and central nervous system (11.5% versus 5.5%), as well as isolation from body fluids of *Candida* spp. (25.1% versus 13.3%), *Acinetobacter* spp. (25.1% versus 4.7%), *Klebsiella* spp. (20.8% versus 7.2%), *Enterococcus* spp. (16.9% versus 8.6%) and *Pseudomonas* spp. (10.9% versus 4.7%). Conversely, a more favourable prognosis was significantly associated with genitourinary (15.5% versus 3.3%) and skin and soft tissue infections (3.3% versus 0.6%) (Table 1, Table 2 and Table 3).

### 2.4. Multivariate Analysis 

Results of the multivariate analysis are reported in Table 4. In regards to characteristics of CA and HA sepsis the results of the multiple logistic regression partly resembled those of the univariate analysis indicating that patients with HA sepsis were significantly more likely to have been admitted from another hospital or ward (Odds Ratio (OR) = 2.92; 95% Confidence Interval (CI) = 1.84–4.63), to have a ≥1 Charlson’s index (OR = 2.21; 95%CI = 1.22–4.01), to be immune suppressed (OR = 1.99; 95%CI = 1.24–3.22), and to have undergone a surgical intervention during hospitalization (OR = 3.74; 95%CI = 2.25–6.21) (Model 1 in Table 4).

Findings from the multivariate logistic regression model revealed that in-hospital death in patients with sepsis was significantly associated with older age (OR = 1.03; 95%CI = 1.01–1.04), admission from another hospital or ward (OR = 2.12; 95%CI = 1.2–3.73), need of haemodialysis (OR = 2.99; 95%CI = 1.4–6.36), and of MV (OR = 11.94; 95%CI = 6.63–21.49), whereas it was less likely in patients with HA sepsis as compared with CA sepsis (OR = 0.53; 95%CI = 0.32–0.9) (Model 2 in Table 4).

## 3. Discussion

This study offers an insight and a thorough description of patients who have been diagnosed a sepsis, as well as the characteristics of their hospitalization and prognosis; moreover it attempted to identify the profiles of patients with CA and HA sepsis and eventual determinants of unfavourable prognosis in the context of a high-income country. The findings of the study, that confirm that CA and HA sepsis generally show distinct clinical characteristics and therefore require tailored management, add considerable knowledge in the recognition of the profile of CA sepsis and in the early identification of in-hospital patients that are at higher risk of developing HA sepsis. 

As expected, the overall picture describing patients with sepsis shows a high frequency of frail subjects affected by one or more comorbidities or immune suppressed, a wide range of primary sites of infection as well as a high number of unrecognized primary sites of infection, whereas most of the isolated micro-organisms are those encountered in HAI, such as *Staphylococcus spp*., *Candida spp*., *Klebsiella spp*., and *Acinetobacter* spp. It should be noted that Rhee et al. [13] have found that the most common causes of suboptimal care in patients with sepsis are delay in antibiotics administration and in source control and inappropriate initial empirical antibiotic therapy. Therefore, according to CDC, among the recommendations for appropriate management of sepsis, a critical role is played by prevention of HAI through appropriate antibiotic prophylaxis, improved awareness of healthcare personnel, and development of effective surveillance methods [14,15].

The rationale for considering CA and HA sepsis as two distinct clinical subsets is justified by several considerations: first of all, while at hospital admission CA sepsis have an already clear diagnosis and are therefore immediately managed as severe diseases, generally in the emergency department, HA sepsis may have a more subtle onset, thus requiring recognition of early markers or characteristics that may represent alerts for implementing adequate therapies; moreover, patients’ hospitalization and prognostic characteristics may differ between the two subsets and a clearer definition of these aspects may be useful to identify specific predictors of HA sepsis and eventual prognostic factors. Page et al. [4] classified patients with sepsis in three groups: (1) patients admitted with a CA sepsis; (2) those who develop sepsis a few days after admission and recognize one or more “healthcare-associated” risk factors prior to admission, and (3) those who are diagnosed a “hospital-acquired” sepsis during hospital stay. Our choice to consider only two groups, merging the “healthcare-associated” and “hospital-acquired” groups into one was driven by the following reasons: first of all, the overall basis of the classification is related to the evaluation of the role of healthcare related factors in the pathogenesis, clinical evolution and prognosis of sepsis, since those factors are common to these two clinical entities, neatly distinguishing them from CA sepsis; moreover, the distinction in only two groups allows a more straightforward analysis reducing complexity in the already complex sepsis scenario; finally, we divided the patients with sepsis according to the classification in three groups, and we found that healthcare-associated sepsis were more similar to the hospital-acquired sepsis as compared to CA sepsis.

The distribution between these two entities in the selected clinical records showed that HA sepsis are somewhat less than half (47.2%) of the diagnosed sepsis. Previous studies have shown frequencies varying from 37% in adult and paediatric patients with sepsis in USA in 2015 [4] to 58% in two studies conducted in 2016 in USA and in 2018 in Brazil [16,17]. However, comparisons may be misleading due to the. different sepsis definitions, as well as different patients’ characteristics. In all studies, indeed, there were no considerable differences in the frequency of these two entities. One of the strength of this study was the implementation of a prediction model that allowed us to suggest that the specific features of HA sepsis as compared to CA sepsis are related to the frailty of patients, that are very likely to be affected by several comorbidities or to be immune suppressed and to the procedures implemented during hospitalization, with a prominent frequency of surgical interventions, MV, and of patients admitted from other hospitals or ward, thus suggesting a prolonged exposure to healthcare procedures. The role of surgical interventions and of specific hospital pathogens has been reported in a previous study [4] and suggests the opportunity to pose attention on patients presenting these predictors for the recognition of early signs of sepsis. 

Mortality associated to sepsis has been investigated in numerous studies that have reported variable estimates ranging from 17% [18] to 55.7% reported in Brazilian ICUs [19]; in-hospital deaths in the present study accounted for one-third of the overall population, but, even for mortality, direct comparisons among studies are difficult to interpret due to the heterogeneity in study designs and patients’ characteristics. When the overall mortality was spread according to sepsis onset, the risk of in-hospital death was 35.4% for HA sepsis and 33.6% for CA sepsis. However, when we modelled mortality, the variables that predicted death, with the exception of older age, were all related to hospital stay, such as admission from another hospital or ward, need of haemodialysis, and need of MV. Once in our model all of these predictors were adjusted for, in-hospital death was significantly more likely in CA as compared to HA sepsis. These considerations may explain the apparently contrasting results with previous studies that found a higher mortality among patients with HA sepsis [4,17]; however in one of these study only crude frequencies were reported and no attempt to identify the independent role of each predictor through adjustment was provided [4], whereas in the other very few healthcare predictors were included in the model [17]. It should be acknowledged that patients admitted with CA sepsis may have developed secondary HA sepsis during their hospital stay. However, clinical records do not always allow to discern symptoms and signs related to CA sepsis to those developed as a secondary HA sepsis; thus, we were not able to perform a subgroup analysis on this CA/HA cohort. Based on this consideration, we cannot exclude that the higher adjusted risk of death among patients with CA sepsis is at least partly the consequence of the presence of this subset of more severe patients within the cohort of those with CA sepsis. It is also worth noting, however, that in a recent study that has investigated prevalence, underlying causes and preventability of sepsis-associated mortality in a cohort of 6 US hospitals, although sepsis was the most common immediate cause of death, the authors reported that the underlying causes of death were severe chronic comorbidities and considered very unlikely that sepsis-associated deaths could be prevented through better hospital care [13]. 

The present analysis has several limitations that need to be addressed. The main limitation of the study is that we used the Angus et al. [20] criteria for sepsis discharge diagnosis, and it is well-known that they are prone to classification bias and to an unclear link between documented infections and organ dysfunction [4]. However, our main aim was to compare HA and CA sepsis, and there is no reason to believe misclassification was not random; thus, we may have had an underestimation of the differences we found between those two conditions. As regards to mortality, due to the retrospective analysis of clinical records, we investigated only in-hospital deaths and no post-discharge surveillance of deaths was available, so underestimation of sepsis-related deaths cannot be excluded. Moreover, it has been reported that the sepsis-3 criteria have a good prognostic accuracy for in-hospital mortality compared to the Angus et al. criteria [21]. Another disadvantage of this study design is that, probably, many different healthcare professionals had been involved in patient care, so the measurement of risk factors and outcomes throughout clinical records might be less accurate and consistent than that achieved with a prospective study design. Finally, the use of administrative data does not allow the measurement of the extent of eventual underreporting of sepsis and the use of clinical records for the evaluation of the characteristics of sepsis is influenced by the frequent incompleteness or incorrectness of this source of data. 

## 4. Materials and Methods

### 4.1. Study Design

A retrospective study was conducted through consultation of clinical records of adult patients with a diagnosis of sepsis. Preliminarily, all hospital discharge records reporting codes for sepsis (995.91), severe sepsis (995.92) or septic shock (785.52) according to the International Classification of Diseases, Ninth Revision, Clinical Modification (ICD-9-CM), and occurred between January 2014 and December 2016 in patients aged 18 and over in the geographic area of the Campania region (Italy) were retrieved from administrative data available from the Campania Regional Health Agency. Subsequently, from the list of 89 hospitals in which discharge diagnoses of sepsis had occurred, a systematic random sample of 42 hospitals was selected and a letter was delivered to the medical directors of these hospitals. The letter explained the purpose of the study and asked permission to access the clinical records of patients diagnosed with a sepsis. Complete anonymity and confidentiality of patients’ data were guaranteed, and the opportunity to participate in the project was emphasized. Following the approval, the selected clinical records were reviewed by three investigators not directly involved in patients’ care and summarized on a standardized data extraction form. The diagnosis of sepsis (formerly called severe sepsis) was defined according to the definition of Angus et al. [20] which includes the occurrence of a serious infection, documented by the ICD-9 codes for sepsis, as well as of organ dysfunctions (respiratory, cardiovascular, neurologic, hematologic, hepatic, renal).

### 4.2. Data Collection Instrument

A structured record form was developed to collect the following characteristics for each patient with a diagnosis of sepsis: socio-demographics characteristics; features regarding hospital admission, stay and discharge (type of admission, admission from another hospital or ward, diagnosis, ward and date of admission and discharge, previous hospitalizations and death); patient characteristics possibly associated with an increased risk of sepsis (cancer, diabetes, systemic steroid use, immune suppressive therapy, poor nutrition, transfusion of blood products, and haemodialysis); Charlson’s et al. [12] comorbidity index; primary site of infection; pathogens isolated from body fluids cultures (blood, urine, cerebrospinal fluid, etc.); procedures associated with an increased risk of sepsis: use of invasive devices [central venous catheter (CVC), urinary catheter (UC), MV] and surgical intervention during hospitalization. 

### 4.3. Classification of Sepsis Cases

Through consultation of clinical records, cases were divided into two groups according to the setting where sepsis originated. A case was classified as CA sepsis if detected on hospital admission or ≤3 days after admission and if there were no healthcare factors in the 30 days preceding hospitalization (i.e., chemotherapy, surgery, presence of CVC, haemodialysis, home healthcare). Cases were classified as HA sepsis if they occurred ≤3 days after hospital admission in presence of healthcare factors in the preceding 30 days or if occurred >3 days after hospital admission. 

The protocol of the study was approved by the Ethical Committee of the University of Campania “Luigi Vanvitelli” of Naples (approval number 409).

### 4.4. Statistical Analysis

The collected data were entered into a database in the form of numeric codes and a data quality check was carried out before starting the statistical analysis. Descriptive analysis summarizing the information for all patients with a diagnosis of sepsis was followed by univariate analysis, that used chi square test or Fischer exact test for all categorical variables and Student’s *t*-test for continuous variables. Then, multivariate stepwise logistic regression models were developed to investigate the independent variables associated with the occurrence of a HA sepsis (no = 0; yes = 1), compared to CA sepsis (Mode1), and those predicting the occurrence of in-hospital death (no = 0; yes = 1) (Model 2). For the identification of variables to be included in the models, our strategy took into account results of the univariate analysis as well as clinical significance of involved factors. Variables that showed a likely association to the outcomes based on the results of the test statistics at the univariate analysis and those which, regardless of the results of the univariate analysis, were judged, based on previous research or on their clinical significance, to have influence on the outcomes, were included in the models.

The following independent variables were included in all models: gender (male = 0, female = 1), age (continuous, in years), Charlson’s index (0 = 0; ≥1 = 1), immune suppressed (no = 0, yes = 1), gastrointestinal (no = 0, yes = 1), bloodstream (no = 0, yes = 1), pulmonary (no = 0, yes = 1), or genitourinary (no = 0, yes = 1) as primary site of infection, *Staphylococcus* spp. (no = 0, yes = 1), *Candida* spp. (no = 0, yes = 1), *Escherichia coli* (no = 0, yes = 1), *Klebsiella* spp. (no = 0, yes = 1), and *Acinetobacter* spp. (no = 0, yes = 1) isolated from body fluids cultures, ward of admission (three categories: ICU = 1, medical wards = 2, surgical wards = 3), admission from another hospital or ward (no = 0; yes = 1), surgical interventions during hospitalization (no = 0, yes = 1), need of haemodialysis (no = 0, yes = 1), need of MV (no = 0, yes= 1), need of ICU (no = 0, yes = 1). The variable length of hospital stay (continuous, in days) was also included in Model 1, and the variable occurrence of a HA sepsis (no = 0; yes = 1) in Model 2.

The significance levels for the exclusion and inclusion of variables in the model were 0.4 and 0.2, respectively. All inferential tests were performed through bilateral hypothesis test with a significance level equal to or less than 0.05. The results of the multivariate regression analyses were reported as ORs and 95% CIs. STATA software version 15 (StataCorp LLC, College Station, TX, USA) was used for the data analysis [22].

## 5. Conclusions

Our study showed that CA and HA sepsis show distinct clinical, prognostic and risk factors profiles and should be managed according to their differential characteristics.

## Figures and Tables

**Table 1 antibiotics-09-00263-t001:** Main characteristics of hospitalized patients with sepsis and univariate analysis with type of sepsis and in-hospital death.

Characteristics	All Patients (*N* = 544)		Community- Acquired Sepsis (*N* = 287)		Healthcare- Associated Sepsis (*N* = 257)		In-Hospital Death (*N* = 183)		Alive and Out of Hospital (*N* = 361)	
			***N***	**(%)**	***N***	**(%)**	***N***	**(%)**	***N***	**(%)**
**Age group, years**	62.8 ± 17.9 (18–97) *		63.2 ± 19 (18–97) *		62.3 ± 16.6 (18–93) *		65.3 ± 15.7 (19–95) *		61.5 ± 18.8 (18–97) *	
			t-test = 0.53, 542 df, *p* = 0.53				t-test = −2.32, 542 df, *p* = 0.02			
**Sex**										
Male	282	(51.8)	149	(51.9)	133	(51.8)	96	(52.5)	186	(51.5)
Female	262	(48.2)	138	(48.1)	124	(48.2)	87	(47.5)	175	(48.5)
			χ2 = 0.002, 1 df, *p* = 0.97				χ2 = 0.043, 1 df, *p* = 0.84			
**Charlson’s index**	3.3 ± 2.5 (0–15) *		3.2 ± 2.6 (0–15) *		3.3 ± 2.4 (0–10) *		3.3 ± 2.5 (0–15) *		3.3 ± 2.5 (0–10) *	
0	71	(13)	47	(16.4)	24	(9.3)	17	(9.3)	54	(15)
≥1	473	(87)	240	(83.6)	233	(90.7)	166	(90.7)	307	(85)
			χ2 = 5.92, 1 df, *p* = 0.015				χ2 = 3.44, 1 df, *p* = 0.06			
**Diabetes**	140	(25.7)	85	(29.6)	55	(21.4)	46	(21.4)	94	(26)
			χ2 = 5.79, 1 df, *p* = 0.029				χ2 = 0.05, 1 df, *p* = 0.82			
**Cancer**	146	(26.8)	50	(17.4)	96	(37.6)	44	(24)	102	(28.3)
			χ2 = 27.4, 1 df, *p* < 0.001				χ2 = 1.1, 1 df, *p* = 0.3			
**Immune suppressed ^+^**	109	(20)	45	(15.7)	64	(24.9)	27	(14.8)	82	(22.7)
			χ2 = 7.2, 1 df, *p* = 0.007				χ2 = 4.8, 1 df, *p* = 0.028			

* Mean ± Standard Deviation (range). ^+^ Subjects with immune suppressive/autoimmune disorders or undergoing immune suppressive drug therapy, radiotherapy, and chemotherapy.

**Table 2 antibiotics-09-00263-t002:** Characteristics of underlying infections (sites and isolated micro-organisms) and univariate analysis with type of sepsis and in-hospital death.

Characteristics	All Patients (*N*= 544)		Community- Acquired Sepsis (*N* = 287)		Healthcare- Associated Sepsis (*N* = 257)		In-Hospital Death (*N* = 183)		Alive and Out of Hospital (*N* = 361)	
			***N***	**(%)**	***N***	**(%)**	***N***	**(%)**	***N***	**(%)**
**Primary site of infection**										
Gastrointestinal ^α^	134	(24.6)	57	(19.9)	77	(30)	43	(23.5)	91	(25.2)
			χ2 = 7.45, 1 df, *p* = 0.006				χ2 = 0.19, 1 df, *p* = 0.66			
Bloodstream	111	(20.4)	56	(19.5)	55	(21.4)	31	(16.9)	80	(22.2)
			χ2 = 0.3, 1 df, *p* = 0.59				χ2 = 2.04, 1 df, *p* = 0.15			
Pulmonary °	97	(17.8)	50	(17.4)	47	(18.3)	61	(33.3)	36	(10)
			χ2 = 0.07, 1 df, *p* = 0.79				χ2 = 45.2, 1 df, *p* < 0.001			
Genitourinary	62	(11.4)	41	(14.3)	21	(8.2)	6	(3.3)	56	(15.5)
			χ2 = 5.02, 1 df, *p* = 0.025				χ2 = 18, 1 df, *p* < 0.001			
Central nervous system	41	(7.6)	31	(10.8)	10	(3.9)	21	(11.5)	20	(5.5)
			χ2 = 9.29, 1 df, *p* = 0.002				χ2 = 6.14, 1 df, *p* = 0.01			
Unknown/Undetermined ^†^	38	(7)	29	(10.1)	9	(3.5)	8	(4.4)	30	(8.3)
			χ2 = 9.1, 1 df, *p* = 0.003				χ2 = 2.9, 1 df, *p* = 0.09			
Cardiovascular	16	(2.9)	8	(2.8)	8	(3.1)	6	(3.3)	10	(2.8)
			χ2 = 0.05, 1 df, *p* = 0.82				χ2 = 0.11, 1 df, *p* = 0.74			
Skin and soft tissue	13	(2.4)	8	(2.8)	5	(2)	1	(0.6)	12	(3.3)
			Fisher exact-test = 0.58, 1 df, *p* = 0.36				Fisher exact-test = 0.07, 1 df, *p* = 0.035			
CVC ^§^-associated infection	12	(2.2)	0	-	12	(4.7)	0	-	12	(3.3)
Surgical site	11	(2)	0	-	11	(4.3)	4	(2.2)	7	(1.9)
							Fisher exact-test = 1.01, 1 df, *p* = 0.54			
Musculoskeletal	6	(1.1)	4	(1.4)	2	(0.8)	1	(0.6)	5	(1.4)
			Fisher exact-test = 0.69, 1 df, *p* = 0.4				Fisher exact-test = 0.67, 1 df, *p* = 0.34			
Disseminated systemic viral	3	(0.6)	3	(1.1)	0	-	1	(0.6)	2	(0.6)
**Type of pathogen** ^b+^										
*Staphylococcus* spp.	135	(24.8)	69	(24)	66	(25.7)	51	(27.9)	84	(23.3)
			χ2 = 0.2, 1 df, *p* = 0.7				χ2 = 1.38, 1 df, *p* = 0.24			
*Candida* spp.	94	(17.3)	37	(12.9)	57	(22.2)	46	(25.1)	48	(13.3)
			χ2 = 8.18, 1 df, *p* = 0.004				χ2 = 11.91, 1 df, *p* = 0.001			
*Escherichia coli*	86	(15.8)	43	(15)	43	(16.7)	24	(13.1)	62	(17.2)
			χ2 = 0.31, 1 df, *p* = 0.58				χ2 = 1.5, 1 df, *p* = 0.22			
*Klebsiella* spp.	64	(11.8)	33	(11.5)	31	(12.1)	38	(20.8)	26	(7.2)
			χ2 = 0.04, 1 df, *p* = 0.84				χ2 = 21.52, 1 df, *p* < 0.001			
*Acinetobacter* spp.	63	(11.5)	25	(8.7)	38	(14.8)	46	(25.1)	17	(4.7)
			χ2 = 4.89, 1 df, *p* = 0.03				χ2 = 49.49, 1 df, *p* < 0.001			
*Enterococcus* spp.	62	(11.4)	27	(9.4)	35	(13.6)	31	(16.9)	31	(8.6)
			χ2 = 2.4, 1 df, *p* = 0.12				χ2 = 8.39, 1 df, *p* = 0.004			
*Pseudomonas* spp.	37	(6.8)	9	(3.2)	28	(10.9)	20	(10.9)	17	(4.7)
			χ2 = 12.9, 1 df, *p* < 0.001				χ2 = 7.41, 1 df, *p* = 0.006			
*Streptococcus* spp.	24	(4.4)	16	(5.6)	8	(3.1)	12	(6.6)	12	(3.3)
			χ2 = 1.9, 1 df, *p* = 0.16				χ2 = 3.01, 1 df, *p* = 0.08			
Unknown/Undetermined	108	(19.9)	41	(16)	67	(23.3)	_	_	_	_
			χ2 = 4.65, 1 df, *p* = 0.031				_			

° Includes upper respiratory infections and pneumonia. ^α^ Includes intra-abdominal, gastrointestinal tract, *C. difficile*, and hepatobiliary infections. † Infection documented by healthcare provider, but source was unknown^. b^ Pathogens isolated from body fluids cultures (blood, urine, cerebrospinal fluid, etc.). + Patients could have more than one pathogen isolated from body fluids culture^. §^ Central Venous Catheter.

**Table 3 antibiotics-09-00263-t003:** Characteristics of patients’ clinical course and outcomes and univariate analysis with type of sepsis and in-hospital death.

Characteristics	All Patients (*N* = 544)		Community- Acquired Sepsis (*N* = 287)		Healthcare- Associated Sepsis (*N* = 257)		In-Hospital Death (*N* = 183)		Alive and Out of Hospital (*N* = 361)	
	**N**	**(%)**	**N**	**(%)**	**N**	**(%)**	**N**	**(%)**	**N**	**(%)**
**Length of hospital stay, days**	20.8 ± 23.5 (1–300) *		17.7 ± 20.1 (1–159) *		24.2 ± 26.4 (1–300) *		21.1 ± 22.2 (1–159) *		20.6 ± 24.1 (1–300) *	
			*t*-test = −3.23, 541 df, *p* = 0.0013				*t*-test = −0.21, 541 df, *p* = 0.83			
**Admitted from another hospital or ward**	210	(38.6)	83	(28.9)	127	(49.4)	113	(61.8)	97	(26.9)
			χ2 = 24, 1 df, *p* =< 0.001				χ2 = 62.3, 1 df, *p* < 0.001			
**Ward of admission**										
Medical	351	(64.5)	212	(73.9)	139	(54.1)	69	(37.7)	284	(78.7)
ICU ^¶^	136	(25)	55	(19.1)	81	(31.5)	95	(51.9)	41	(11.4)
Surgical	57	(10.5)	20	(7)	37	(14.4)	19	(10.4)	36	(10.5)
			χ2 = 23.6, 2 df, *p* < 0.001				χ2 = 110.6, 2 df, *p* < 0.001			
**Need of MV ^β^**	218	(40.1)	99	(34.5)	119	(46.3)	151	(82.5)	67	(18.6)
			χ2 = 7.87, 1 df, *p* = 0.005				χ2 = 206.83, 1 df, *p* < 0.001			
**Need of ICU** ^¶^	94	(17.3)	52	(18.1)	42	(16.3)	59	(32.2)	35	(9.7)
			χ2 = 0.3, 1 df, *p* = 0.584				χ2 = 43.18, 1 df, *p* < 0.001			
**Need of haemodialysis**	73	(13.4)	33	(11.5)	40	(15.6)	50	(27.3)	23	(6.4)
			χ2 = 1.93, 1 df, *p* = 0.17				χ2 = 45.9, 1 df, *p* < 0.001			
**In-Hospital mortality**	183	(33.6)	92	(32.1)	91	(35.4)	-	-	-	-
			χ2 = 0.68, 1 df, *p* = 0.41							
**Surgery during hospitalization**	143	(26.3)	39	(13.6)	104	(40.5)	55	(30.1)	88	(24.4)
			χ2 = 50.6, 1 df, *p* < 0.001				χ2 = 2.02, 1 df, *p* = 0.16			
**Surgical wound classification** ^+^										
Clean	46	(32.2)	11	(28.2)	35	(33.7)	10	(18.2)	36	(40.9)
Clean-contaminated	97	(67.8)	28	(71.8)	69	(66.3)	45	(81.8)	52	(59.1)
			χ2 = 0.39, 1 df, *p* = 0.53				χ2 = 8.01, 1 df, *p* = 0.005			
**Sepsis**										
Community-acquired	287	(52.8)	-	-	-	-	92	(32.1)	195	(67.9)
Healthcare-associated sepsis	257	(47.2)	-	-	-	-	91	(35.4)	166	(64.6)
							χ2 = 0.68, 1 df, *p* = 0.41			

* Mean ± SD (range). ^¶^ Intensive Care Unit. ^β^ Mechanical ventilation. ^+^ Only among patients who have undergone surgical intervention during hospitalization (*N* = 143).

**Table 4 antibiotics-09-00263-t004:** Multivariate logistic regression analyses to characterize factors associated with occurrence of HA^+^ sepsis and in-hospital mortality related to sepsis.

Variable	OR ^σ^	SE ^τ^	95%CI ^φ^	*p*-Value
**Model 1.** Occurrence of a HA^+^ sepsis (sample size = 543) ^ω^				
Log likelihood = −319.21, χ^2^ = 112.78 (14 df), *p* < 0.0001				
**Charlson’s index ≥ 1**	2.21	0.67	1.22–4.01	0.009
**Immune suppressed**	1.99	0.49	1.24–3.22	0.005
**Admission from another hospital or ward**	2.92	0.69	1.84–4.63	<0.001
**Surgical intervention during hospitalization**	3.74	0.97	2.25–6.21	<0.001
**Gastrointestinal primary site of infection**	1.38	0.34	0.84–2.24	0.201
**Bloodstream primary site of infection**	1.62	0.4	0.99–2.64	0.051
***Candida*** ***spp.*** **^α^**	1.56	0.42	0.92–2.65	0.101
***Klebsiella*** **spp.** **^α^**	0.62	0.21	0.31–1.21	0.161
***Acinetobacter*** **spp.** **^α^**	1.47	0.53	0.72–2.98	0.289
**Ward of admission**				
ICU °*	1.0			
Medical	1.93	1.37	0.48–7.8	0.354
Surgical	3.39	2.65	0.73–15.68	0.117
**Length of hospital stay, days**	1.01	0.01	0.99–1.01	0.117
**Need of MV ^β^**	2.05	1.51	0.48–8.68	0.330
**Need of ICU** ***°***	0.3	0.23	0.06–1.37	0.120
**Model 2.** Occurrence of in-hospital death (sample size = 544) ^ω^				
Log likelihood = −210.87, χ^2^ = 273.07 (12 df), *p* < 0.0001				
**Age, years (continuous)**	1.03	0.01	1.01–1.04	<0.001
**Admission from another hospital or ward**	2.12	0.61	1.2–3.73	0.009
**Need of haemodialysis**	2.99	1.15	1.4–6.36	0.005
**Need of MV ^β^**	11.94	3.58	6.63–21.49	<0.001
**Patient with HA *^+^* sepsis**	0.53	0.14	0.32–0.9	0.019
**Pulmonary primary site of infection**	1.5	0.48	0.8–2.82	0.206
**Genitourinary primary site of infection**	0.35	0.2	0.11–1.07	0.065
***Acinetobacter*** **spp.** **^α^**	1.85	0.81	0.78–4.37	0.163
***Klebsiella*** **spp.** **^α^**	1.6	0.7	0.68–3.78	0.283
**Ward of admission**				
ICU °*	1.0			
Medical	0.7	0.23	0.37–1.32	0.268
Surgical	1.92	0.89	0.78–4.74	0.157

^σ^ Odds Ratio. ^τ^ Standard Error. ^φ^ Confidence Interval. ^ω^ Variables deleted by backward elimination procedure were not included in the table. ^+^ Healthcare-associated sepsis. *Reference category. ^α^ Pathogen isolated from body fluids cultures (blood, urine, cerebrospinal fluid, etc.). ^β^ Mechanical ventilation. ° Intensive Care Unit.

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
