# Peer review of "Community-Acquired and Healthcare-Associated Sepsis: Characteristics and in-Hospital Mortality in Italy"

_antibiotics, 2020, doi:10.3390/antibiotics9050263_

Round 1

Reviewer 1 Report

The article, “Community-acquired and healthcare-associated sepsis: characteristics and in-hospital mortality in Italy,” by Giuseppe et al. is a retrospective review of 544 cases of sepsis in the Campania region of Italy comparing community-acquired sepsis to  healthcare-associated sepsis. This group has performed a thorough review of a cohort of hospitalized septic patients. Interestingly, they found that patients with community-acquired sepsis had a higher likelihood of death than those with healthcare-associated sepsis.

Major Comments

  1. In their introduction, the authors divide sepsis into three established groups: community-acquired, healthcare-associated, and hospital-acquired. I would recommend using this same framework for their study rather than community-acquired vs healthcare-associated. If this is not possible, the reason for not dividing into these three categories should be explicitly described.
  2. The tables used to convey their reported data are very difficult to follow. To improve clarity, I would recommend splitting Table 1 into several separate tables – one with general patient characteristics, one with infection characteristics (surgical wounds, site of infection, and isolated microbe), and a third with clinical outcomes. Clinical outcomes would include length of hospital stay, need for ICU level of care, need for mechanical ventilation, and mortality. I would also recommend the authors expand their evaluation of clinical outcomes to include development of ARDS, acute liver injury/failure, acute kidney injury, and initiation of hemodialysis (if not previously ESRD) to provide further insight into clinically relevant outcomes for these patients.
  3. Clinically, many patients are admitted with sepsis and develop secondary infections (sepsis) during their hospital stay. This group does not describe how they categorized patients initially admitted with CA sepsis who went on to develop secondary HA sepsis. If these patients were included in HA sepsis, this would under-represent the severity of CA sepsis. It's plausible that if CA sepsis patients who went on to develop secondary infections were indeed classified into CA sepsis, then this could explain their higher adjusted risk of death. If so, I would recommend performing a subgroup analysis on this CA/HA cohort.

Minor Comments

  1. I would recommend including length of hospital stay broken down by CA vs HA sepsis.
  2. I would recommend sub-dividing the results section into separate headings including: general patient characteristics, clinical outcomes, univariate analysis, and multivariate analysis.
  3. It would be helpful to report the number of patients without an identified organism for their infection divided by CA vs HA.
  4. Recommend significant edits to this manuscript to improve clarity. Notable grammar and spelling mistakes throughout the manuscript.

Author Response

Response to Reviewer 1 Comments

Major Comments

Point 1: In their introduction, the authors divide sepsis into three established groups: community-acquired, healthcare-associated, and hospital-acquired. I would recommend using this same framework for their study rather than community-acquired vs healthcare-associated. If this is not possible, the reason for not dividing into these three categories should be explicitly described.

Response 1: (Lines 244-255) In response to Point 1 we agree with the reviewer that a different classification of sepsis is reported in the introduction as compared to the one we chose in the paper. Our choice was driven by the following reasons: 1) we have also evaluated the patients with sepsis according to the classification in three groups, and we found that healthcare-associated sepsis were more similar as phenotype to the hospital-acquired sepsis; 2) the overall basis of the classification is related to the verification of the role of healthcare related factors in the pathogenesis, clinical evolution and prognosis of sepsis; 3) the distinction in only two phenotypes would provide a more straightforward analysis reducing complexity in the already complex sepsis scenario. To avoid misunderstanding, we have now motivated our choice in the Discussion section.

Point 2: The tables used to convey their reported data are very difficult to follow. To improve clarity, I would recommend splitting Table 1 into several separate tables – one with general patient characteristics, one with infection characteristics (surgical wounds, site of infection, and isolated microbe), and a third with clinical outcomes. Clinical outcomes would include length of hospital stay, need for ICU level of care, need for mechanical ventilation, and mortality. I would also recommend the authors expand their evaluation of clinical outcomes to include development of ARDS, acute liver injury/failure, acute kidney injury, and initiation of haemodialysis (if not previously ESRD) to provide further insight into clinically relevant outcomes for these patients.

Response 2: (Lines 89; 122; 137) As suggested, Table 1 has now been split in 3 Tables and we have provided more information on clinical outcomes (length of hospital stay, need for ICU level of care, need for mechanical ventilation, and need of haemodialysis), whereas clinical records were not considered reliable for the extraction of data on development of ARDS, acute liver injury and kidney injury. New variables have also been included in the models.

Point 3: Clinically, many patients are admitted with sepsis and develop secondary infections (sepsis) during their hospital stay. This group does not describe how they categorized patients initially admitted with CA sepsis who went on to develop secondary HA sepsis. If these patients were included in HA sepsis, this would under-represent the severity of CA sepsis. It is plausible that if CA sepsis patients who went on to develop secondary infections were indeed classified into CA sepsis, then this could explain their higher adjusted risk of death. If so, I would recommend performing a subgroup analysis on this CA/HA cohort.

Response 3: (Lines 308-312) In response to this point, we agree with the reviewer that CA sepsis may have developed in HA sepsis, and since we classified CA sepsis based on the time of occurrence, we agree that it is plausible that we have classified as CA sepsis also those that went on to develop HA secondary infections. However, our source of data (clinical record) does not always allow to discern symptoms and signs related to the sepsis acquired in the community to those developed as a secondary HA sepsis; therefore we are not able to perform a subgroup analysis on this CA/HA cohort. Therefore, since we agree with the reviewer that this could, at least partly, explain the finding of a higher adjusted risk of death of CA sepsis, we have acknowledged this point in the limitations of the study.

Minor Comments

Point 1: I would recommend including length of hospital stay broken down by CA vs HA sepsis.

Response 2: (Table 3) As suggested, length of hospital stay broken down by CA vs HA sepsis has been included.

Point 2: I would recommend sub-dividing the results section into separate headings including: general patient characteristics, clinical outcomes, univariate analysis, and multivariate analysis.

Response 2: (Lines 80; 117; 149; 180) As suggested the results section has been sub-divided into separate headings including: general patient characteristics, clinical outcomes, univariate analysis, and multivariate analysis.

Point 3: It would be helpful to report the number of patients without an identified organism for their infection divided by CA vs HA.

Response 3: (Table 2) As suggested, number of patients without an identified organism for their infection divided by CA vs HA has been reported.

Point 4: Recommend significant edits to this manuscript to improve clarity. Notable grammar and spelling mistakes throughout the manuscript.

Response 4: As suggested, the manuscript has undergone editing for the improvement of language and clarity.

Response to Reviewer 1 Comments

Major Comments

Point 1: In their introduction, the authors divide sepsis into three established groups: community-acquired, healthcare-associated, and hospital-acquired. I would recommend using this same framework for their study rather than community-acquired vs healthcare-associated. If this is not possible, the reason for not dividing into these three categories should be explicitly described.

Response 1: (Lines 244-255) In response to Point 1 we agree with the reviewer that a different classification of sepsis is reported in the introduction as compared to the one we chose in the paper. Our choice was driven by the following reasons: 1) we have also evaluated the patients with sepsis according to the classification in three groups, and we found that healthcare-associated sepsis were more similar as phenotype to the hospital-acquired sepsis; 2) the overall basis of the classification is related to the verification of the role of healthcare related factors in the pathogenesis, clinical evolution and prognosis of sepsis; 3) the distinction in only two phenotypes would provide a more straightforward analysis reducing complexity in the already complex sepsis scenario. To avoid misunderstanding, we have now motivated our choice in the Discussion section.

Point 2: The tables used to convey their reported data are very difficult to follow. To improve clarity, I would recommend splitting Table 1 into several separate tables – one with general patient characteristics, one with infection characteristics (surgical wounds, site of infection, and isolated microbe), and a third with clinical outcomes. Clinical outcomes would include length of hospital stay, need for ICU level of care, need for mechanical ventilation, and mortality. I would also recommend the authors expand their evaluation of clinical outcomes to include development of ARDS, acute liver injury/failure, acute kidney injury, and initiation of haemodialysis (if not previously ESRD) to provide further insight into clinically relevant outcomes for these patients.

Response 2: (Lines 89; 122; 137) As suggested, Table 1 has now been split in 3 Tables and we have provided more information on clinical outcomes (length of hospital stay, need for ICU level of care, need for mechanical ventilation, and need of haemodialysis), whereas clinical records were not considered reliable for the extraction of data on development of ARDS, acute liver injury and kidney injury. New variables have also been included in the models.

Point 3: Clinically, many patients are admitted with sepsis and develop secondary infections (sepsis) during their hospital stay. This group does not describe how they categorized patients initially admitted with CA sepsis who went on to develop secondary HA sepsis. If these patients were included in HA sepsis, this would under-represent the severity of CA sepsis. It is plausible that if CA sepsis patients who went on to develop secondary infections were indeed classified into CA sepsis, then this could explain their higher adjusted risk of death. If so, I would recommend performing a subgroup analysis on this CA/HA cohort.

Response 3: (Lines 308-312) In response to this point, we agree with the reviewer that CA sepsis may have developed in HA sepsis, and since we classified CA sepsis based on the time of occurrence, we agree that it is plausible that we have classified as CA sepsis also those that went on to develop HA secondary infections. However, our source of data (clinical record) does not always allow to discern symptoms and signs related to the sepsis acquired in the community to those developed as a secondary HA sepsis; therefore we are not able to perform a subgroup analysis on this CA/HA cohort. Therefore, since we agree with the reviewer that this could, at least partly, explain the finding of a higher adjusted risk of death of CA sepsis, we have acknowledged this point in the limitations of the study.

Minor Comments

Point 1: I would recommend including length of hospital stay broken down by CA vs HA sepsis.

Response 2: (Table 3) As suggested, length of hospital stay broken down by CA vs HA sepsis has been included.

Point 2: I would recommend sub-dividing the results section into separate headings including: general patient characteristics, clinical outcomes, univariate analysis, and multivariate analysis.

Response 2: (Lines 80; 117; 149; 180) As suggested the results section has been sub-divided into separate headings including: general patient characteristics, clinical outcomes, univariate analysis, and multivariate analysis.

Point 3: It would be helpful to report the number of patients without an identified organism for their infection divided by CA vs HA.

Response 3: (Table 2) As suggested, number of patients without an identified organism for their infection divided by CA vs HA has been reported.

Point 4: Recommend significant edits to this manuscript to improve clarity. Notable grammar and spelling mistakes throughout the manuscript.

Response 4: As suggested, the manuscript has undergone editing for the improvement of language and clarity.

Reviewer 2 Report

The study deals with a challenging and evergreen issue, sepsis, and is an attempt to tackle the complexity of sepsis clinical phenotypes by distinguishing two clinical profiles, defined by the setting of occurrence: the community or the healthcare settings. These profiles are analyzed in details and factors predicting poor prognosis are evaluated.

Specific Comments:

In the Introduction, sepsis were categorized into three different groups, but analysis and results refer to two groups: community-acquired and healthcare-associated sepsis. Please clarify this point.

No mention has been made on the characteristics of the enrolled hospitals in the study. Please provide. Moreover, did you verify whether participating hospitals were different from non-participating ones?

Add the abbreviation healthcare-associated (HA) sepsis when it first appears in the text (results section) and delete it in the methods section.

It is not clear how the variables to be included in the models were chosen. The model building strategy should be explained.

The importance of prevention activities in sepsis management, such as early diagnosis and appropriateness of antibiotic prophylaxis, must be emphasized in the Discussion section.

The limitation due to the retrospective design of the study should be expanded in the Discussion section.

Table 1 contains too much data and is hard to read. It could be split in two tables.

Please, check consistencies between text and Tables. For instance, “whereas it was less likely in patients admitted in medical wards compared to ICU (OR=1.02; 95%CI=1.01-1.03)”: this OR and CI are different from those reported in the Table. Also check abbreviations in the Table, that are not consistent with the text.

Model 1 in Table 2: no data is reported for surgical wards: why?

Please, check all the text for typos and language.

Author Response

Response to Reviewer 2 Comments

Specific Comments

Point 1: In the Introduction, sepsis was categorized into three different groups, but analysis and results refer to two groups: community-acquired and healthcare-associated sepsis. Please clarify this point.

Response 1: (Lines 244-255) In response to Point 1 we agree with the reviewer that a different classification of sepsis is reported in the introduction as compared to the one we chose in the paper. Our choice was driven by the following reasons: 1) we have also evaluated the patients with sepsis according to the classification in three groups, and we found that healthcare-associated sepsis were more similar as phenotype to the hospital-acquired sepsis; 2) the overall basis of the classification is related to the verification of the role of healthcare related factors in the pathogenesis, clinical evolution and prognosis of sepsis; 3) the distinction in only two phenotypes would provide a more straightforward analysis reducing complexity in the already complex sepsis scenario. To avoid misunderstanding, we have now motivated our choice in the Discussion section.

Point 2: No mention has been made on the characteristics of the enrolled hospitals in the study. Please provide. Moreover, did you verify whether participating hospitals were different from non-participating ones?

Response 2: (Lines 74-78) As suggested, characteristics of hospitals have been included and we have reported that no substantial differences were found between participating and non-participating hospitals.

Point 3: Add the abbreviation healthcare-associated (HA) sepsis when it first appears in the text (results section) and delete it in the methods section.

Response 3: As suggested, the abbreviation has been included when it first appeared (results section).

Point 4: It is not clear how the variables to be included in the models were chosen. The model building strategy should be explained.

Response 4: (Lines 369-374) As suggested, the model building strategy has been included in the statistical methods section.

Point 5: The importance of prevention activities in sepsis management, such as early diagnosis and appropriateness of antibiotic prophylaxis, must be emphasized in the Discussion section.

Response 5: (Lines 231-237) As suggested, a more detailed Discussion on the importance of prevention activities in sepsis management has been included.

Point 6: The limitation due to the retrospective design of the study should be expanded in the Discussion section.

Response 6: (Lines 309-312) As suggested, limits of the retrospective design have been expanded.

Point 7: Table 1 contains too much data and is hard to read. It could be split in two tables.

Response 7: (Lines 89; 122; 137) As suggested, and according also to Reviewer 1, Table 1 has been split into 3 Tables.

Point 8: Please, check consistencies between text and Tables. For instance, “whereas it was less likely in patients admitted in medical wards compared to ICU (OR=1.02; 95%CI=1.01-1.03)”: this OR and CI are different from those reported in the Table. Also check abbreviations in the Table, that are not consistent with the text.

Response 8: As suggested, several inconsistencies between the text and the Tables have been checked and corrected.

Point 9: Model 1 in Table 2: no data is reported for surgical wards: why?

Response 9: (Table 4) As suggested, Model 1 has been corrected as regard to data on surgical wards (it was the sub-heading Ward of admission).

Point 10: Please, check all the text for typos and language.

Response 10: As suggested, the manuscript has undergone editing for the improvement of language and clarity.

Round 2

Reviewer 1 Report

The revision of “Community-acquired and healthcare-associated sepsis: characteristics and in-hospital mortality in Italy,” by Di Giuseppe G et al. is improved with the integration of previous critiques. This paper provides some insight into the clinical characteristics of patients in Italy with community-acquired versus healthcare-associated sepsis.

Major Comments

None. The authors have addressed my major concerns.

Minor Comments

  • In Results, page 2, lines 69-71, would recommend including interquartile range rather than absolute range.
  • Would begin a new paragraph at line 79, page 2 in the results section beginning with “At the Univariate Analysis…” Additionally, I would recommend modifying “At the Univariate Analysis,” to “In univariate analysis…”
  • In results, page 2, line 84 would recommend changing “in these patients” to “in patients with HA sepsis.”
  • Modify Table 1 Title to include “Univariate Analysis” in the title.
  • In Table 1, please clarify why Length of Hospital Stay was not broken down by community-acquired and health-care acquired sepsis. If no specific reason for this omission, please include. If omitted, rationale for this omission should be included in the methods section.
  • In table 1, “Cancer” section has a p-value of 0001. Assume this should read < 0.001, but please adjust accordingly.
  • In table 1, please modify “Immunosuppression status” to Immune-suppressed and include an annotation describing what categories of patients are included in this status.
  • In Table 1, “Surgical Wound Classification” section, it seems that the clean-contaminated and “No” sections are reversed. I would also recommend removing the no surgery patients from this section completely.
  • In Discussion, page 10, line 136, would recommend removing “Phenotype” and change to “distinct clinical characteristics,” as phenotype implies a specific pathophysiologic profile, which would likely sub-divide HA and CA sepsis.
  • In Discussion section, page 10, line 136, would remove the words “probably demand,” and insert “require.” This is already an accepted practice pattern. Antibiotics are tailored to location of sepsis onset.
  • On page 10, line 146, please clarify the meaning of “Emergency Ward,” as this is not common terminology in all countries.
  • On page 11, line 197, would add “or incorrect” after “incomplete.”
  • The Conclusions section are not clear to me. I think the findings of this study are descriptive. They suggest that data, like those included in this study, should be taken into account when developing treatment protocols for community-acquire and healthcare-associated sepsis. I’m not sure what the authors mean when they say “…that a major contribution to early detection and prognosis of sepsis may be provided by the consideration that CA and HA sepsis have to be managed according to their differential characteristics…”

Author Response

Minor Comments

Point 1: In Results, page 2, lines 69-71, would recommend including interquartile range rather than absolute range.

Response 1: (Lines 75-76) As suggested, we have added interquartile range.

Point 2: Would begin a new paragraph at line 79, page 2 in the results section beginning with “At the Univariate Analysis…” Additionally, I would recommend modifying “At the Univariate Analysis,” to “In univariate analysis…”

Response 2: (Lines 151, 158) As suggested, a new paragraph has been included in the results section beginning with “At the Univariate Analysis…” and “At the Univariate Analysis,” has been modified as “In univariate analysis…”.

Point 3: In results, page 2, line 84 would recommend changing “in these patients” to “in patients with HA sepsis.”

Response 3: (Line 154) As suggested, in the results section we have modified “in these patients” to “in patients with HA sepsis.”

Point 4: Modify Table 1 Title to include “Univariate Analysis” in the title.

Response 4: As suggested, we have modified the title in Tables 1, 2 and 3.

Point 5: In Table 1, please clarify why Length of Hospital Stay was not broken down by community-acquired and health-care acquired sepsis. If no specific reason for this omission, please include. If omitted, rationale for this omission should be included in the methods section.

Response 5: In response to your request, in the first revised version we have already broken-down Length of Hospital stay by community-acquired and health-care acquired sepsis, and, as of your suggestion, it now appears in Table 3.

Point 6: In table 1, “Cancer” section has a p-value of 0001. Assume this should read < 0.001, but please adjust accordingly.

Response 6: There was a typing error. As suggested, we have modified “<0001” with “< 0.001”.

Point 7: In table 1, please modify “Immunosuppression status” to Immune-suppressed and include an annotation describing what categories of patients are included in this status.

Response 7: As suggested, in Table 1 we have modified “Immunosuppression status” with Immune-suppressed and we have added a footnote that describes the categories of patients included. This change has been reported all over the text.

Point 8: In Table 1, “Surgical Wound Classification” section, it seems that the clean-contaminated and “No” sections are reversed. I would also recommend removing the no surgery patients from this section completely.

Response 8: As suggested, we have removed the “no surgery” patients from the section on “Surgical Wound Classification”, that in the revised version is in Table 3.

Point 9: In Discussion, page 10, line 136, would recommend removing “Phenotype” and change to “distinct clinical characteristics,” as phenotype implies a specific pathophysiologic profile, which would likely sub-divide HA and CA sepsis.

Response 9: (Lines 27, 42, 202) As suggested, we have modified in the Discussion section and also in the Abstract and Introduction section “phenotype” with “distinct clinical characteristics”.

Point 10: In Discussion section, page 10, line 136, would remove the words “probably demand,” and insert “require.” This is already an accepted practice pattern. Antibiotics are tailored to location of sepsis onset.

Response 10: (Line 202) As suggested, in the Discussion section we have modified “demand” with “require”.

Point 11: On page 10, line 146, please clarify the meaning of “Emergency Ward,” as this is not common terminology in all countries.

Response 11: (Line 218) In response to this point, as suggested we have modified “emergency ward” with “emergency department”, to better specify what we meant.

Point 12: On page 11, line 197, would add “or incorrect” after “incomplete.”

Response 12: (Line 288) As suggested, “or incorrectness” has been added after “incompleteness.”

Point 13: The Conclusions section are not clear to me. I think the findings of this study are descriptive. They suggest that data, like those included in this study, should be taken into account when developing treatment protocols for community-acquire and healthcare-associated sepsis. I’m not sure what the authors mean when they say “…that a major contribution to early detection and prognosis of sepsis may be provided by the consideration that CA and HA sepsis have to be managed according to their differential characteristics…”

Response 13: (Line 359-360) In response to this point, our suggestion, sustained by the results of the study, was that the decisions on the management of sepsis should also take into account whether they are CA or HA sepsis, since their clinical and prognostic characteristics are different. To avoid misunderstanding, we have rephrased the conclusions according to your suggestion.

Minor Comments

Point 1: In Results, page 2, lines 69-71, would recommend including interquartile range rather than absolute range.

Response 1: (Lines 75-76) As suggested, we have added interquartile range.

Point 2: Would begin a new paragraph at line 79, page 2 in the results section beginning with “At the Univariate Analysis…” Additionally, I would recommend modifying “At the Univariate Analysis,” to “In univariate analysis…”

Response 2: (Lines 151, 158) As suggested, a new paragraph has been included in the results section beginning with “At the Univariate Analysis…” and “At the Univariate Analysis,” has been modified as “In univariate analysis…”.

Point 3: In results, page 2, line 84 would recommend changing “in these patients” to “in patients with HA sepsis.”

Response 3: (Line 154) As suggested, in the results section we have modified “in these patients” to “in patients with HA sepsis.”

Point 4: Modify Table 1 Title to include “Univariate Analysis” in the title.

Response 4: As suggested, we have modified the title in Tables 1, 2 and 3.

Point 5: In Table 1, please clarify why Length of Hospital Stay was not broken down by community-acquired and health-care acquired sepsis. If no specific reason for this omission, please include. If omitted, rationale for this omission should be included in the methods section.

Response 5: In response to your request, in the first revised version we have already broken-down Length of Hospital stay by community-acquired and health-care acquired sepsis, and, as of your suggestion, it now appears in Table 3.

Point 6: In table 1, “Cancer” section has a p-value of 0001. Assume this should read < 0.001, but please adjust accordingly.

Response 6: There was a typing error. As suggested, we have modified “<0001” with “< 0.001”.

Point 7: In table 1, please modify “Immunosuppression status” to Immune-suppressed and include an annotation describing what categories of patients are included in this status.

Response 7: As suggested, in Table 1 we have modified “Immunosuppression status” with Immune-suppressed and we have added a footnote that describes the categories of patients included. This change has been reported all over the text.

Point 8: In Table 1, “Surgical Wound Classification” section, it seems that the clean-contaminated and “No” sections are reversed. I would also recommend removing the no surgery patients from this section completely.

Response 8: As suggested, we have removed the “no surgery” patients from the section on “Surgical Wound Classification”, that in the revised version is in Table 3.

Point 9: In Discussion, page 10, line 136, would recommend removing “Phenotype” and change to “distinct clinical characteristics,” as phenotype implies a specific pathophysiologic profile, which would likely sub-divide HA and CA sepsis.

Response 9: (Lines 27, 42, 202) As suggested, we have modified in the Discussion section and also in the Abstract and Introduction section “phenotype” with “distinct clinical characteristics”.

Point 10: In Discussion section, page 10, line 136, would remove the words “probably demand,” and insert “require.” This is already an accepted practice pattern. Antibiotics are tailored to location of sepsis onset.

Response 10: (Line 202) As suggested, in the Discussion section we have modified “demand” with “require”.

Point 11: On page 10, line 146, please clarify the meaning of “Emergency Ward,” as this is not common terminology in all countries.

Response 11: (Line 218) In response to this point, as suggested we have modified “emergency ward” with “emergency department”, to better specify what we meant.

Point 12: On page 11, line 197, would add “or incorrect” after “incomplete.”

Response 12: (Line 288) As suggested, “or incorrectness” has been added after “incompleteness.”

Point 13: The Conclusions section are not clear to me. I think the findings of this study are descriptive. They suggest that data, like those included in this study, should be taken into account when developing treatment protocols for community-acquire and healthcare-associated sepsis. I’m not sure what the authors mean when they say “…that a major contribution to early detection and prognosis of sepsis may be provided by the consideration that CA and HA sepsis have to be managed according to their differential characteristics…”

Response 13: (Line 359-360) In response to this point, our suggestion, sustained by the results of the study, was that the decisions on the management of sepsis should also take into account whether they are CA or HA sepsis, since their clinical and prognostic characteristics are different. To avoid misunderstanding, we have rephrased the conclusions according to your suggestion.

Minor Comments

Point 1: In Results, page 2, lines 69-71, would recommend including interquartile range rather than absolute range.

Response 1: (Lines 75-76) As suggested, we have added interquartile range.

Point 2: Would begin a new paragraph at line 79, page 2 in the results section beginning with “At the Univariate Analysis…” Additionally, I would recommend modifying “At the Univariate Analysis,” to “In univariate analysis…”

Response 2: (Lines 151, 158) As suggested, a new paragraph has been included in the results section beginning with “At the Univariate Analysis…” and “At the Univariate Analysis,” has been modified as “In univariate analysis…”.

Point 3: In results, page 2, line 84 would recommend changing “in these patients” to “in patients with HA sepsis.”

Response 3: (Line 154) As suggested, in the results section we have modified “in these patients” to “in patients with HA sepsis.”

Point 4: Modify Table 1 Title to include “Univariate Analysis” in the title.

Response 4: As suggested, we have modified the title in Tables 1, 2 and 3.

Point 5: In Table 1, please clarify why Length of Hospital Stay was not broken down by community-acquired and health-care acquired sepsis. If no specific reason for this omission, please include. If omitted, rationale for this omission should be included in the methods section.

Response 5: In response to your request, in the first revised version we have already broken-down Length of Hospital stay by community-acquired and health-care acquired sepsis, and, as of your suggestion, it now appears in Table 3.

Point 6: In table 1, “Cancer” section has a p-value of 0001. Assume this should read < 0.001, but please adjust accordingly.

Response 6: There was a typing error. As suggested, we have modified “<0001” with “< 0.001”.

Point 7: In table 1, please modify “Immunosuppression status” to Immune-suppressed and include an annotation describing what categories of patients are included in this status.

Response 7: As suggested, in Table 1 we have modified “Immunosuppression status” with Immune-suppressed and we have added a footnote that describes the categories of patients included. This change has been reported all over the text.

Point 8: In Table 1, “Surgical Wound Classification” section, it seems that the clean-contaminated and “No” sections are reversed. I would also recommend removing the no surgery patients from this section completely.

Response 8: As suggested, we have removed the “no surgery” patients from the section on “Surgical Wound Classification”, that in the revised version is in Table 3.

Point 9: In Discussion, page 10, line 136, would recommend removing “Phenotype” and change to “distinct clinical characteristics,” as phenotype implies a specific pathophysiologic profile, which would likely sub-divide HA and CA sepsis.

Response 9: (Lines 27, 42, 202) As suggested, we have modified in the Discussion section and also in the Abstract and Introduction section “phenotype” with “distinct clinical characteristics”.

Point 10: In Discussion section, page 10, line 136, would remove the words “probably demand,” and insert “require.” This is already an accepted practice pattern. Antibiotics are tailored to location of sepsis onset.

Response 10: (Line 202) As suggested, in the Discussion section we have modified “demand” with “require”.

Point 11: On page 10, line 146, please clarify the meaning of “Emergency Ward,” as this is not common terminology in all countries.

Response 11: (Line 218) In response to this point, as suggested we have modified “emergency ward” with “emergency department”, to better specify what we meant.

Point 12: On page 11, line 197, would add “or incorrect” after “incomplete.”

Response 12: (Line 288) As suggested, “or incorrectness” has been added after “incompleteness.”

Point 13: The Conclusions section are not clear to me. I think the findings of this study are descriptive. They suggest that data, like those included in this study, should be taken into account when developing treatment protocols for community-acquire and healthcare-associated sepsis. I’m not sure what the authors mean when they say “…that a major contribution to early detection and prognosis of sepsis may be provided by the consideration that CA and HA sepsis have to be managed according to their differential characteristics…”

Response 13: (Line 359-360) In response to this point, our suggestion, sustained by the results of the study, was that the decisions on the management of sepsis should also take into account whether they are CA or HA sepsis, since their clinical and prognostic characteristics are different. To avoid misunderstanding, we have rephrased the conclusions according to your suggestion.

Minor Comments

Point 1: In Results, page 2, lines 69-71, would recommend including interquartile range rather than absolute range.

Response 1: (Lines 75-76) As suggested, we have added interquartile range.

Point 2: Would begin a new paragraph at line 79, page 2 in the results section beginning with “At the Univariate Analysis…” Additionally, I would recommend modifying “At the Univariate Analysis,” to “In univariate analysis…”

Response 2: (Lines 151, 158) As suggested, a new paragraph has been included in the results section beginning with “At the Univariate Analysis…” and “At the Univariate Analysis,” has been modified as “In univariate analysis…”.

Point 3: In results, page 2, line 84 would recommend changing “in these patients” to “in patients with HA sepsis.”

Response 3: (Line 154) As suggested, in the results section we have modified “in these patients” to “in patients with HA sepsis.”

Point 4: Modify Table 1 Title to include “Univariate Analysis” in the title.

Response 4: As suggested, we have modified the title in Tables 1, 2 and 3.

Point 5: In Table 1, please clarify why Length of Hospital Stay was not broken down by community-acquired and health-care acquired sepsis. If no specific reason for this omission, please include. If omitted, rationale for this omission should be included in the methods section.

Response 5: In response to your request, in the first revised version we have already broken-down Length of Hospital stay by community-acquired and health-care acquired sepsis, and, as of your suggestion, it now appears in Table 3.

Point 6: In table 1, “Cancer” section has a p-value of 0001. Assume this should read < 0.001, but please adjust accordingly.

Response 6: There was a typing error. As suggested, we have modified “<0001” with “< 0.001”.

Point 7: In table 1, please modify “Immunosuppression status” to Immune-suppressed and include an annotation describing what categories of patients are included in this status.

Response 7: As suggested, in Table 1 we have modified “Immunosuppression status” with Immune-suppressed and we have added a footnote that describes the categories of patients included. This change has been reported all over the text.

Point 8: In Table 1, “Surgical Wound Classification” section, it seems that the clean-contaminated and “No” sections are reversed. I would also recommend removing the no surgery patients from this section completely.

Response 8: As suggested, we have removed the “no surgery” patients from the section on “Surgical Wound Classification”, that in the revised version is in Table 3.

Point 9: In Discussion, page 10, line 136, would recommend removing “Phenotype” and change to “distinct clinical characteristics,” as phenotype implies a specific pathophysiologic profile, which would likely sub-divide HA and CA sepsis.

Response 9: (Lines 27, 42, 202) As suggested, we have modified in the Discussion section and also in the Abstract and Introduction section “phenotype” with “distinct clinical characteristics”.

Point 10: In Discussion section, page 10, line 136, would remove the words “probably demand,” and insert “require.” This is already an accepted practice pattern. Antibiotics are tailored to location of sepsis onset.

Response 10: (Line 202) As suggested, in the Discussion section we have modified “demand” with “require”.

Point 11: On page 10, line 146, please clarify the meaning of “Emergency Ward,” as this is not common terminology in all countries.

Response 11: (Line 218) In response to this point, as suggested we have modified “emergency ward” with “emergency department”, to better specify what we meant.

Point 12: On page 11, line 197, would add “or incorrect” after “incomplete.”

Response 12: (Line 288) As suggested, “or incorrectness” has been added after “incompleteness.”

Point 13: The Conclusions section are not clear to me. I think the findings of this study are descriptive. They suggest that data, like those included in this study, should be taken into account when developing treatment protocols for community-acquire and healthcare-associated sepsis. I’m not sure what the authors mean when they say “…that a major contribution to early detection and prognosis of sepsis may be provided by the consideration that CA and HA sepsis have to be managed according to their differential characteristics…”

Response 13: (Line 359-360) In response to this point, our suggestion, sustained by the results of the study, was that the decisions on the management of sepsis should also take into account whether they are CA or HA sepsis, since their clinical and prognostic characteristics are different. To avoid misunderstanding, we have rephrased the conclusions according to your suggestion.
